# Sexual health behavior, health status, and knowledge among queer womxn and trans men in Kenya: An online cross-sectional study

**Stephanie Haase** [ORCID]*, **Alex Müller, Virginia Zweigenthal**

School of Public Health and Family Medicine, University of Cape Town, Cape Town, South Africa

* hsxste003@myuct.ac.za

## Abstract

### Introduction

Little research has been conducted on the sexual health needs and risk behaviors of queer womxn and trans men, making it difficult to identify specific health needs and disparities. This is especially the case in the Global South, where their needs are poorly understood. This study presents findings on demographics, sources of information, sexual (risk) behaviors, and substance use in Kenyan queer womxn and trans men.

### Methods

An online survey among 335 Kenyan queer womxn and trans men was used to collect data on sexual health, risk behavior, health information sources, and substance use. The participants needed to have had at least one self-identified female sexual partner.

### Results

The sample presented young, highly-educated queer womxn and trans men. A high incidence of childhood sexual trauma found was found. Risk behaviors included sexual activities with partners of multiple genders, violence, and low use of barrier methods. One in three participants had been treated for an STD in the previous year. The incidences of smoking and drinking were high, and a quarter of participants indicated having taken drugs at least once a month or more. The internet was either the first or second most important source of sexual health information for 44.1% of the participants, followed by schools (30.9%).

### Discussion and conclusion

Our findings indicate that queer womxn and trans men are at risk of negative sexual health outcomes due to a lack of appropriate information, risk behavior, substance use, and low uptake of sexual health services. Kenya's Penal Code still criminalizes consensual same-sex activities and may play a role in perpetuating barriers that prohibit them from making healthier choices. Developing tailored programming and policies require local, national, and global stakeholders to engage with the inclusion of queer womxn and trans men's sexual health needs within strategic planning and healthcare delivery.

**Data Availability Statement:** Due to the sensitive nature of the dataset, data will not be made publicly available. Some data can be made available from the corresponding author on reasonable request.

The University of Cape Town Human Research Ethics Committee can also be reached for data requests. hrec-enquiries@uct.ac.za.

**Funding:** The authors received no specific funding for this work.

**Competing interests:** The authors have declared that no competing interests exist.

# Introduction

Queer womxn–participants preferred this explicitly inclusive intersectional feminist designation–and trans men are a neglected population regarding sexual health education, service provision, and research [1, 2], making it difficult to identify specific disparities [1, 3]. Research emerged only recently on their specific clinical needs and health inequalities [4, 5]. Disparities originate from synergetic structural, interpersonal, and psychological factors [6–8] and leave them at risk of negative health outcomes [4, 9, 10]. The available research suggests that queer womxn and trans men have a high burden of disease regarding sexually transmitted diseases and infections (STD/STIs) and HIV, STI-related cancers, as well as mental health concerns and violence [11]. In particular, transgender men's unique needs are poorly understood [12].

Risk behavior, such as drug use and incidental sex with men, can leave queer womxn and trans men at risk of contracting STIs [4, 13, 14], even though it is often erroneously assumed that they have a low risk [9]. Other risky sexual behaviors include concurrent substance use, lack of contraceptive and barrier method use, a high number of lifetime and recent sexual partners, and low age at first intercourse [15]. A lack of information on safe sex, experiences of violence, and a lack of appropriate services can leave queer womxn at risk of negative health outcomes [10]. According to a 2015 US study, lesbian womxn are more likely to engage in behaviors that could leave them at risk of STIs and HIV/AIDS compared to their heterosexual peers [9].

Alcohol consumption and binge drinking were more prevalent in female-identifying sexual minorities between 20 and 34 than in their heterosexual peers [16], and lesbian and bisexual study participants were more likely to have alcohol-related problems such as arguments or legal problems [17]. These attitudes can affect physical and mental health and lead to risky sexual behavior [18–20].

Most of the existing health research on sexual and gender minorities is conducted in resource-rich settings, leaving the Global South underrepresented [21]. The lack of understanding of the health disparities is especially evident in countries where people could face serious human rights violations based on their actual or perceived sexual orientation, gender identity, and expression (SOGIE) [22]. In many African countries, sexual activities between consenting adults of the same sex and or gender are illegal [23]. Laws criminalizing consensual same-sex activities are often ill-defined and applied more broadly than to consensual same-sex activities only, restricting availability and access to sexual health information or services [23] or violating human rights, such as forced anal examinations and STI tests used as evidence in suspected cases of consensual same-sex activity [24, 25].

In Kenya, where the Penal Code criminalizes consensual same-sex, there are health care programs that target men having sex with men (MSM), a key population in the fight against HIV/AIDS [21]. This focus on MSM, however, leaves queer womxn and trans men underserved and neglected. Their needs are poorly understood; only two studies [10, 26] exclusively focus on them and explicitly address their sexual health, and one study examines queer refugee womxn [27]. By comparison, there are at least 53 publications on MSM. As the articles had limited scope, it is assumed that there is a lack of information on sexual health and risk behavior which could negatively affect healthcare resource utilization and health status, especially in countries such as Kenya, where consensual same-sex activities are criminalized, but also elsewhere.

A lack of sexual health education may leave queer womxn and trans men poorly equipped to cope with unique circumstances and pressures different than the ones cisgender women face, a barrier that inclusive sexual health curricula could address [28]. Despite the Kenyan government initiating efforts to include comprehensive sexuality education (CSE) in primary

school curricula, the existing messaging continues to be hetero- and cis-normative, conservative, not rights-based, and focuses on abstinence and HIV awareness [29].

This article aims at assessing queer womxn and trans men's overall self-reported health status, sexual health and risk behaviors, knowledge around sexual health, and information seeking, and measures how sexual health behavior relates to several environmental covariates in a restrictive legal environment.

## Methods

All study protocols were approved by the Health Sciences Human Research Ethics Committee of the University of Cape Town (HREC 033/2019) on March 28th, 2019 and the Amref Kenya ESRC (P659-2019) on July 11th, 2019. Consent was obtained in written format.

A cross-sectional online survey among a snowballing-recruited sample of Kenyan queer womxn and trans men was used to collect data around sexual health, risk behavior, sources of information, and substance use.

### Eligibility

To be eligible to participate in the survey, respondents had to have been assigned female biological sex at birth, have had at least one female sexual partner (self-identified or identified/perceived as female by the participant), and participated in consensual same-sex activity in the past three years. Consensual same-sex activity was defined as any act of physical contact including at least two people with the intent to create sexual pleasure for at least one of the partaking individuals, such as vaginal and anal stimulation, oral and penetrative sex (digital, sex toys), and mutual masturbation. Participants needed to have been born and spent most of their lives in Kenya. Participants included those identifying as heterosexual, lesbian, bisexual, lesbian, gay, and queer cisgender womxn, gender-non-conforming people (assigned female gender at birth), and trans men.

### Participant recruitment

Data were collected through an online survey. A pilot was designed to ensure that the survey was culturally and language-appropriate and shared with 'influencers' to enable buy-in. Influencers were key figures in the community known to the researchers; trusted by other queer womxn and trans men. It was assumed that influencers would be more likely to share the survey if they approved of its content and, through pilot participation, felt a sense of ownership. The survey link was distributed to pilot participants; three of whom then shared the survey within the queer womxn and trans men communities, mainly via WhatsApp groups.

Data were collected over three weeks. Participants were compensated for their internet usage with a 200 Kenyan Shillings (approx. US$1.80) mobile data voucher after completing the survey. To discourage participants from answering the survey multiple times, they stated that they would participate only once on the informed consent form. A question on the survey asked how many times a participant had completed it; only participants who said they had not previously responded could proceed.

### Outcome measures

The survey consisted of previously validated, self-administered tools for sexual health behavior, health, and substance use. Table 1 gives an overview of the instruments used.

The tool to assess sexual risk behavior was an abbreviated version of the Safe Sex Behavior Questionnaire [30]. Further isolated, relevant questions from the Youth Risk Behavior Surveillance System [33], the Behavioral Risk Factor Surveillance System [34], and the Illustrative

**Table 1. Indicators, tools, and outcomes.**

| Indicator | Tool | Outcomes |
|---|---|---|
| Sexual health behavior | Safe Sex Behavior questionnaire (abbreviated) [30] | Eleven statements on safe sex and risky behavior. The scores range from 11 to 44, with lower scores an indication for engaging in risky behavior and difficulties discussing and negotiating safe sex with partners |
| Health 1 | PROMIS: physical health [31] | Two overall mental and physical health statements ranked on a 5-point Likert scale ranging from 1 to 5: poor to excellent. |
| Health 2 | PROMIS mental health [31] | Two statements on mental health ranked on a 5-point Likert scale ranging from 1 to 5: poor to excellent. |
| Substance use | Substance use [32] | Seven statements on how many days over the past month participants used various substances. The scores range from 7 to 35, with higher scores indicating higher levels of and or more frequent substance use. |

Questionnaire for Interview Surveys with Young People [35] were chosen to examine other areas of sexual health, such as sources of information, age at first sexual activity, number of life-time sexual partners, anal sex, and testing for HIV, STD/STIs and PAP-smears. PROMIS scales were used for overall mental and physical health [31]. To measure substance use, a tool originally created by the CDC, adapted for use in LGBT populations, was used [32].

## Demographic data and covariates

Sociodemographic data were collected, including age, sexual orientation, gender identity, education, geographics, frequency of attendance of religious services, and socioeconomic status. Table 2 gives an overview of participants' questions and answer choices for sexual orientation and gender identity.

## Data collection

REDCap [36], a secure software for collecting, storing, and managing scientific data, was utilized. Web-based sampling was used for a purposive sample of survey participants. Additionally, snowballing was employed–participants were issued a link they could share with potential

**Table 2. Sexual orientation and gender identity survey choices.**

| Question | Answer choices |
|---|---|
| How do you identify in terms of sexual orientation (to whom you are sexually, emotionally, and physically attracted)? Please tick one. | Lesbian (womxn mostly attracted to womxn) <br> Bisexual (attracted mostly to men and womxn) <br> Gay (man mostly attracted to men) <br> Homosexual (attracted mostly to people of your gender) <br> Heterosexual/straight (attracted mostly to people of the opposite gender) <br> Asexual (not sexually attracted to people) <br> Queer (not heterosexual/straight) <br> Other |
| In terms of gender identity, how do you identify? Please tick one. | Female <br> Male <br> Trans woman (assigned male biological sex at birth, identifies as female) <br> Trans man (assigned female biological sex at birth, identifies as male) <br> Gender non-conforming (behavior and expression do not conform with traditional, male, and female gender norms) <br> Don't know <br> Other |

other participants. Another publication describes the sampling method in detail *(anonymized for peer review, under review)*.

As there are no data available on the size of this population, sample size estimation for regression models was used, since linear regression models were utilized to determine which individual, relationship, community, and societal stressors influenced sexual health decision-making [37]. As there are 14 variables, a minimum sample size of 140 was required. This number was doubled to increase power.

A total of 360 participants responded to the survey; 335 were included. Twenty-five participants were excluded for not meeting the inclusion criteria (8 participants) or not completing the informed consent section (17 participants).

## Data analysis

To assess and report correlations, the REDCap data were imported into SPSS 26 [38]. Basic descriptive statistics for the demographics were presented as percentages, means, and standard deviations. Scores were calculated for all previously validated tools.

The association of sexual health behavior (dependent variable, outcome) and gender identity, sexual orientation, age, education level, geographics, frequency of attendance of religious services, relationship status, and socioeconomic status (independent variables) were evaluated independently using linear regression models. The independent variables for frequency of religious attendance, sexual orientation, gender identity, and education were logically grouped to increase the number of participants in each group, and hence power. Table 3 shows the groupings.

Descriptive analyses of the major variables were conducted to illustrate the distribution of the variables in question; if not normally distributed, log-transformations were performed to be able to conduct univariate linear regression analyses. For categorical variables used later in the analyses, dummy variables were created [37]. Univariate linear regression analyses were performed using SPSS 26. Only associations between the dependent and independent variables with significant p-values ($p < 0.05$) were reported.

## Results

### Demographics

Participants from 32 of the 47 Kenyan counties responded, with the majority coming from areas of high population density–Nairobi, Kiambu, Kisumu, Nakuru, and Mombasa counties. The majority (60.9%, n = 204) indicated living in a town (see Fig 1). Overall, the participants represented a young sample, with the majority (44.9%, n = 150) between 25 and 34 years old (see Fig 2).

More than half of the respondents (56.2%, n = 189) had either attended some university or completed their university education (see Fig 3). The majority (29.9%, n = 100) said they were currently students, while 19.4% were formally employed (see Fig 4). Over half of the respondents (57%, n = 191) said they did not have enough money to cover their basic needs like food and housing.

The frequency of religious service attendance was high, with 24.9% (n = 83) attending either daily or several times a week (see Fig 5).

Most participants (82.8%, n = 278) identified as womxn and 72.7% (n = 247) as lesbian (see Figs 6 and 7).

### Health

**Physical and mental health.**   When asked about their health, the majority (33.2%, n = 111) rated their physical health as good, and 41.5% (n = 139) said their activities of daily

**Table 3. Covariates and new groupings.**

| Covariate | Previous grouping | New Grouping |
|---|---|---|
| Frequency of religious attendance | Daily (n = 34) | Often (24.7%) |
| | Several times a week (n = 49) | |
| | Once/week (n = 69) | Frequently (36%) |
| | Several times per month (n = 52) | |
| | Once a month (n = 91) | Rarely (39.3%) |
| | Rarely (n = 30) | |
| | Never (n = 9) | |
| Sexual orientation | Lesbian (n = 242) | Lesbian(73.3%) |
| | Homosexual/Gay (n = 5) | |
| | Bisexual (n = 42) | Bisexual (12.5%) |
| | Queer (n = 40) | Queer (13.8%) |
| | Other (n = 1) | |
| | Heterosexual (n = 5) | |
| Gender identity | Womxn/woman (n = 277) | Cisgender (82.9%) |
| | Other (n = 1) | |
| | Transgender (n = 27) | Transgender (8%) |
| | Gender-non-conforming (n = 29) | Gender-non-conforming (9.1%) |
| Education | No school (n = 1) | Primary school (excluded from analysis) |
| | Primary school (n = 2) | |
| | Some secondary school (n = 26) | Secondary school (28.3%) |
| | Secondary school (n = 69) | |
| | Vocational training (n = 48) | Vocational (14.3%) |
| | Some university education (n = 104) | University (57.4%) |
| | Completed university (n = 84) | |

living (walking, climbing stairs, carrying groceries, moving a chair) were very good. Two (1.2%) of the respondents indicated that they were living with a disability (not further specified). Many (39.4%, n = 132) said their mental health was good, as did over a third (33.8%, n = 119) regarding their satisfaction with social activities and relationships. Fig 8 gives an overview of the different health indicators.

## Sexual health

**PAP smear.** The majority (74.3%, n = 249) had never had a PAP smear. Of the 25.7% (n = 86) who had one, 43% (n = 37) had a PAP smear within the past year, 23.3% (n = 20) within the past two years, 16.3% (n = 14) within the past three years, 14% (n = 12) within the past five years and 3.5% (n = 3) had had one longer than five years ago.

**HIV and STD testing.** Most respondents (76.1%, n = 255) said they had been previously tested for HIV, but 0.6% (n = 2) were unsure. Of the ones that had been tested, 76.9% (n = 196) had been tested in the past year, 12.5% (n = 39) within the past two years, 6.7% (n = 17) within the past three years, 0.8% (n = 2) within the past five years and 0.4% (n = 1) had had one longer than five years ago.

Within the past year, 35.5% (n = 206) said they had been tested for an STD/STI other than HIV; 3.1% (n = 10) were unsure. A third (33%, n = 110) had been treated for an STD/STI, 1.3% (n = 4) were not sure. 67.5% (n = 74) said their partner was also tested and treated for the same STD/STI.

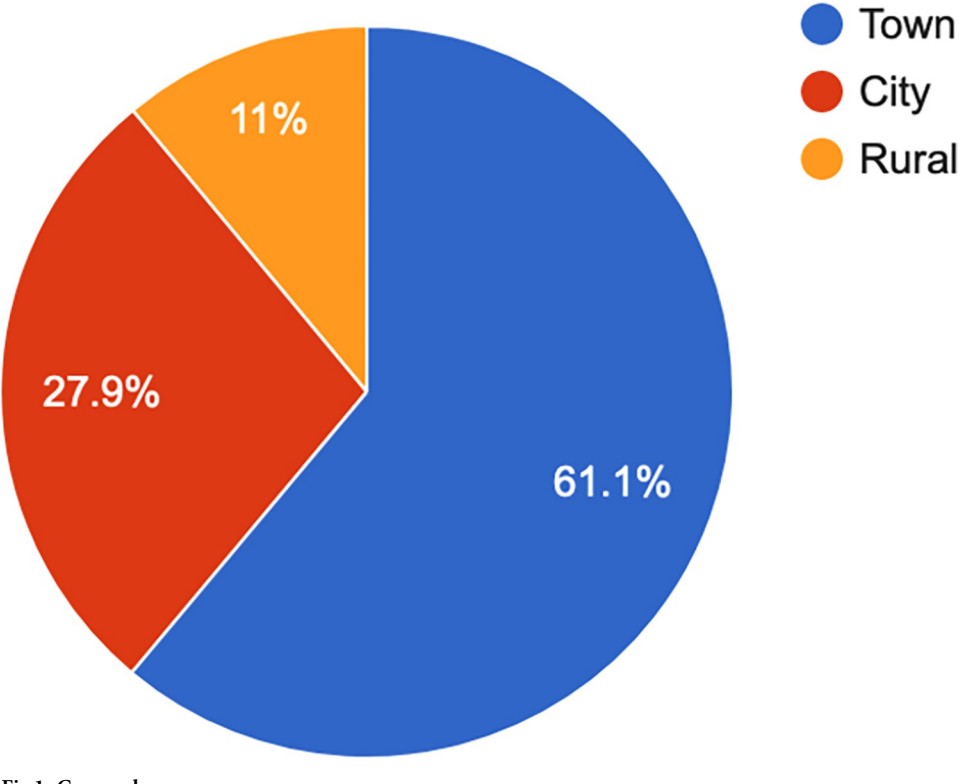

**Fig 1. Geography.**

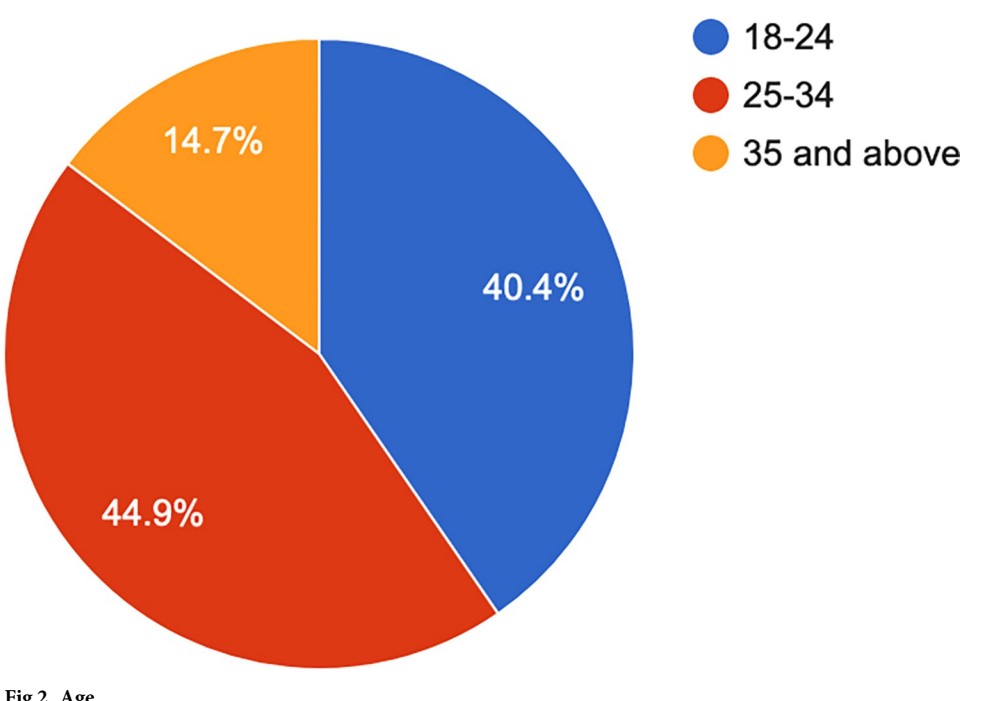

**Fig 2. Age.**

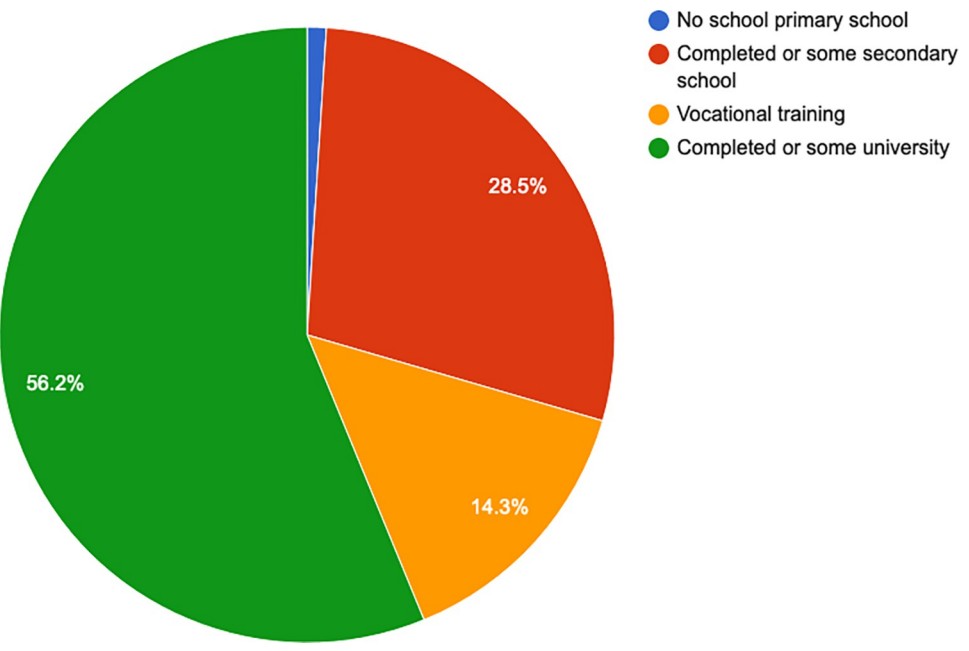

**Fig 3. Education.**

## Contraception

The majority, 69.3% (n = 232), had never used any form of contraception. For those who had, the most used methods were male condoms (66.3%, n = 67) and emergency contraception (62%,

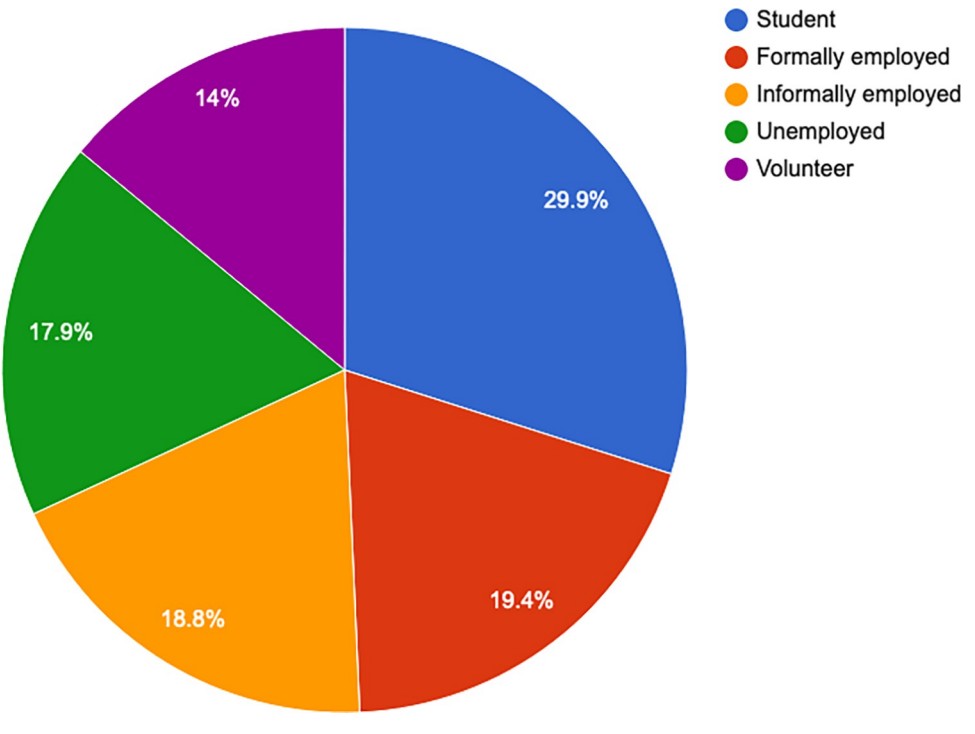

**Fig 4. Employment status.**

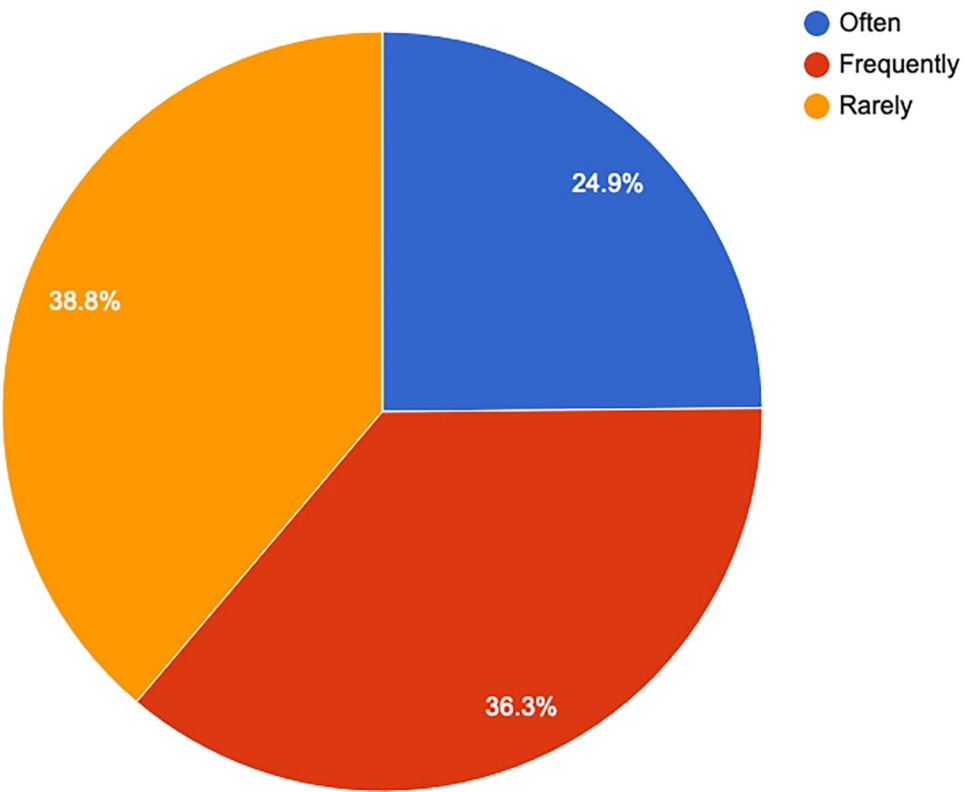

**Fig 5. Religiosity/frequency of attending religious services.**

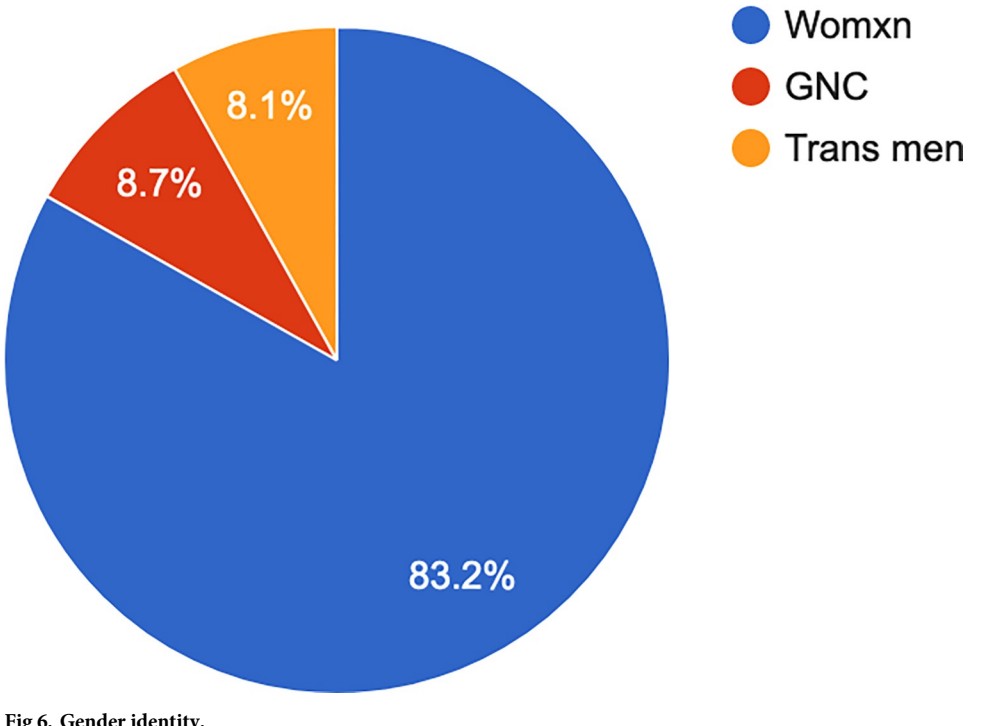

**Fig 6. Gender identity.**

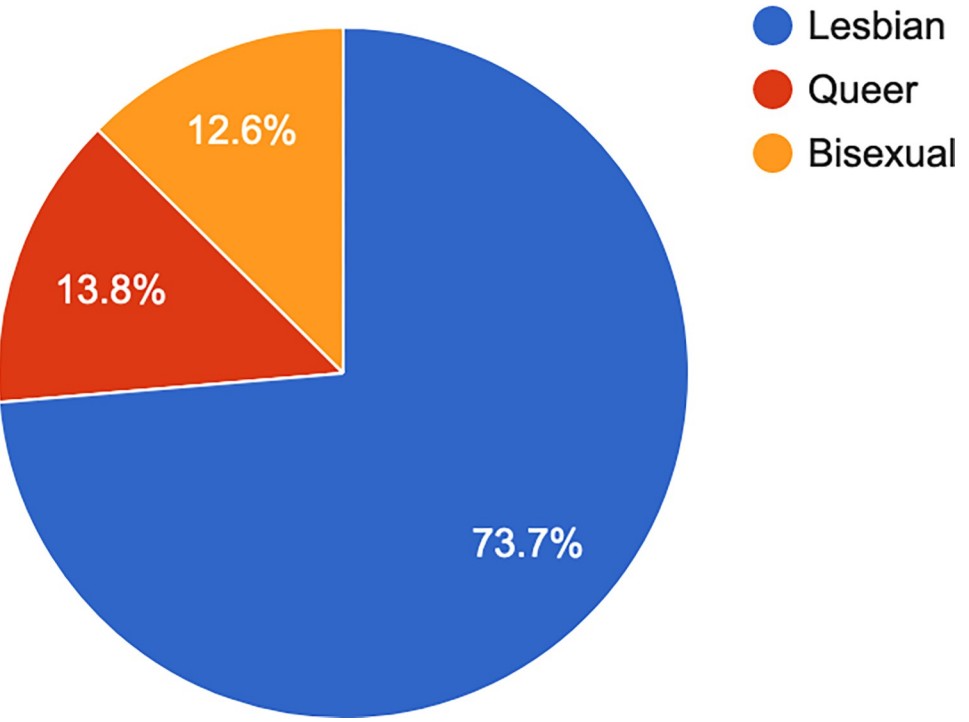

**Fig 7. Sexual orientation.**

n = 62). As a reason for using birth control, 20.3% (n = 20) said they used it to prevent unintended pregnancies, 16.7% (n = 17) to prevent STD/STIs, and 3.3% (n = 3) for medical reasons.

## Pregnancy

Of the respondents, 78.8% (n = 271) had never been pregnant, 9.9% (n = 33) had an abortion, 9.3% (n = 31) have a child or children and 2% (n = 7) had a miscarriage.

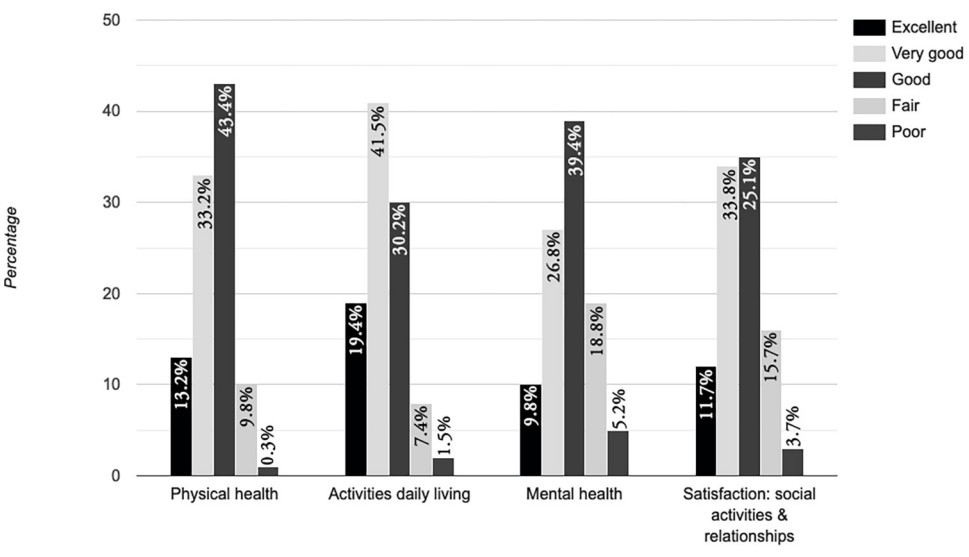

**Fig 8. Health status.**

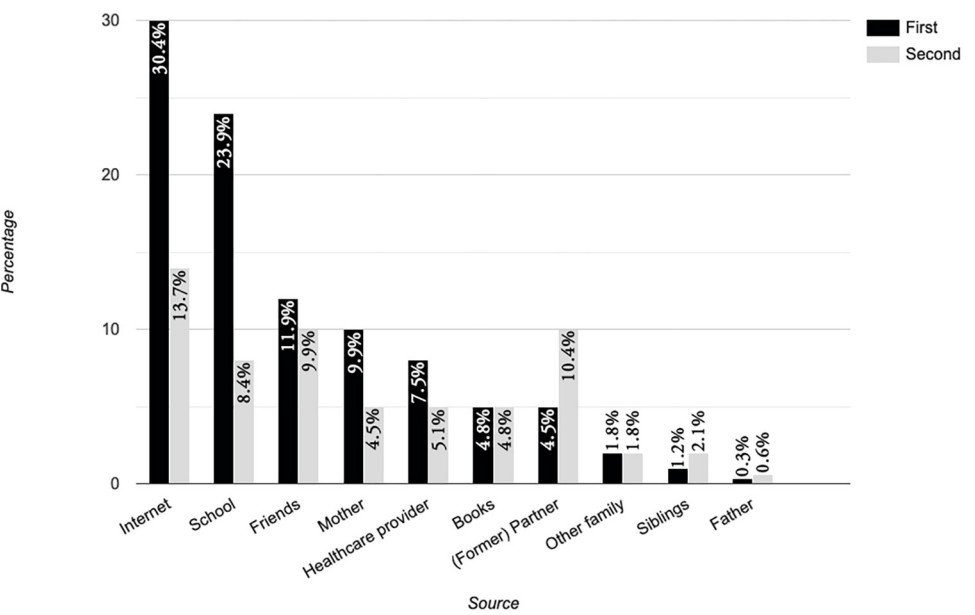

**Fig 9. Sources of information.**

## SRH knowledge and Comprehensive Sexuality Education (CSE)

A quarter (25.2%, n = 84) did not know some of the signs and symptoms of STIs, and only a fifth (20.6%, n = 69) knew that some STIs could be without symptoms. Half of the respondents correctly identified ulcers and sores in the genital area and vaginal discharge as symptoms of STIs.

For 44.1% (n = 148), the internet was either the first or second most important source of SRHR information, followed by school (30.9%, n = 109) and friends (20.9%, n = 73); see Fig 9. The majority (72.3%, n = 242) said they had received some form of sexual health education in school, but 10.2% (n = 34) were unsure.

## Sexual practices

The mean age of first-time sex was 17.3 years, with a minimum of 3 years and a maximum of 21 (SD, 4.1). Regression showed that GNC participants had sex on average 1.7 years earlier than the cis-gender participants (17.6–1.7; p < 0.05; CI 95% -3.3 –-0.21). Of the participants, 14.2% indicated having been younger than 12; the youngest was three years old.

The mean number of lifetime sex partners was 12.9 (SD, 21.6; range:1–221). Trans participants had significantly more lifetime sex partners than the cis-gender participants (11.1 + 22.9; p < 0.05; CI 95% 14.6–31.3). In the past three months, the range of number of sex partners was from 0 to 10, with a mean of 1.42 (SD, 1.2).

## Heterosexual interactions and sex partners

Of the respondents, 28.1% (n = 94) said that they had sex with men at least once in their lifetime. We did not ask about the frequency of heterosexual interactions. The most frequently mentioned sex partners were cisgender women (91.6%, n = 307) and cisgender men (28.1%, n = 94) (see Table 4).

**Table 4. Gender identities of lifetime sex partners.**

| Identity | n | % |
|---|---|---|
| Cis-gender women | 307 | 91.6 |
| Cis-gender men | 94 | 28.1 |
| GNC people (female biological sex assigned at birth) | 45 | 13.4 |
| Trans men with gender-affirming surgery | 7 | 2.1 |
| Other | 7 | 2.1 |
| Trans women without gender-affirming surgery | 6 | 1.8 |
| Trans men without gender-affirming surgery | 5 | 1.6 |
| GNC people (male biological sex assigned at birth) | 5 | 1.5 |
| Intersex people | 3 | 0.9 |
| Trans women with gender-affirming surgery | 2 | 0.6 |

## Relationship status

The majority (57.5%, n = 153) said they were dating or in a relationship with one person or were currently not dating or in a relationship (24%, n = 82). See Fig 10 for more details.

## Anal play

Of the 24.2% (n = 81) of respondents who said they had engaged in anal play within the past year, involving hands, fingers, tongue, or mouth (77.8%, n = 63), sex toys (13.6%, n = 11) or penises (8.6% = 7).

## Transactional sex

Of the respondents, 14.6% (n = 49) said they had ever given or received money, drugs, or gifts for sex.

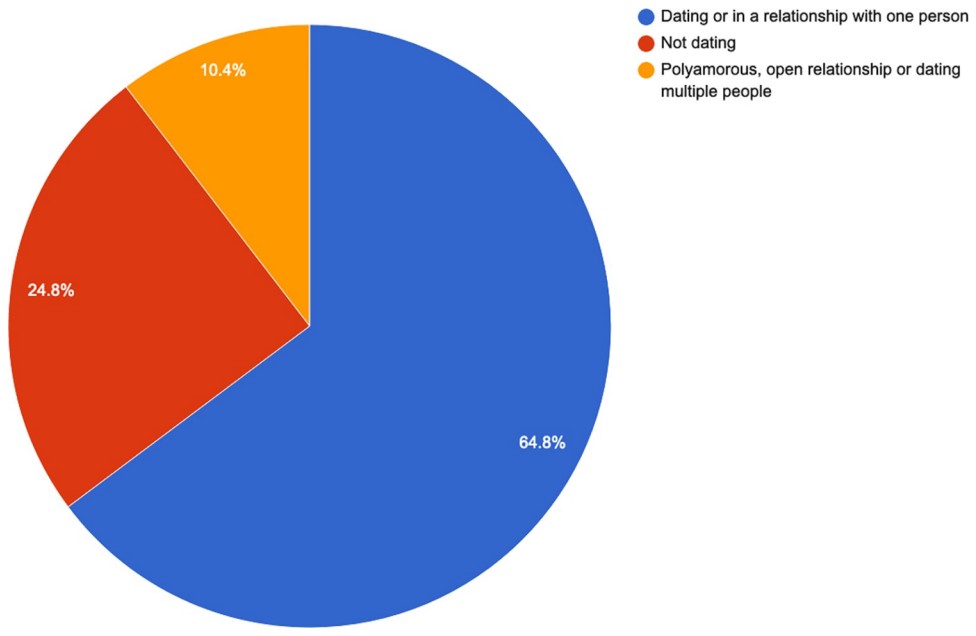

**Fig 10. Relationship status.**

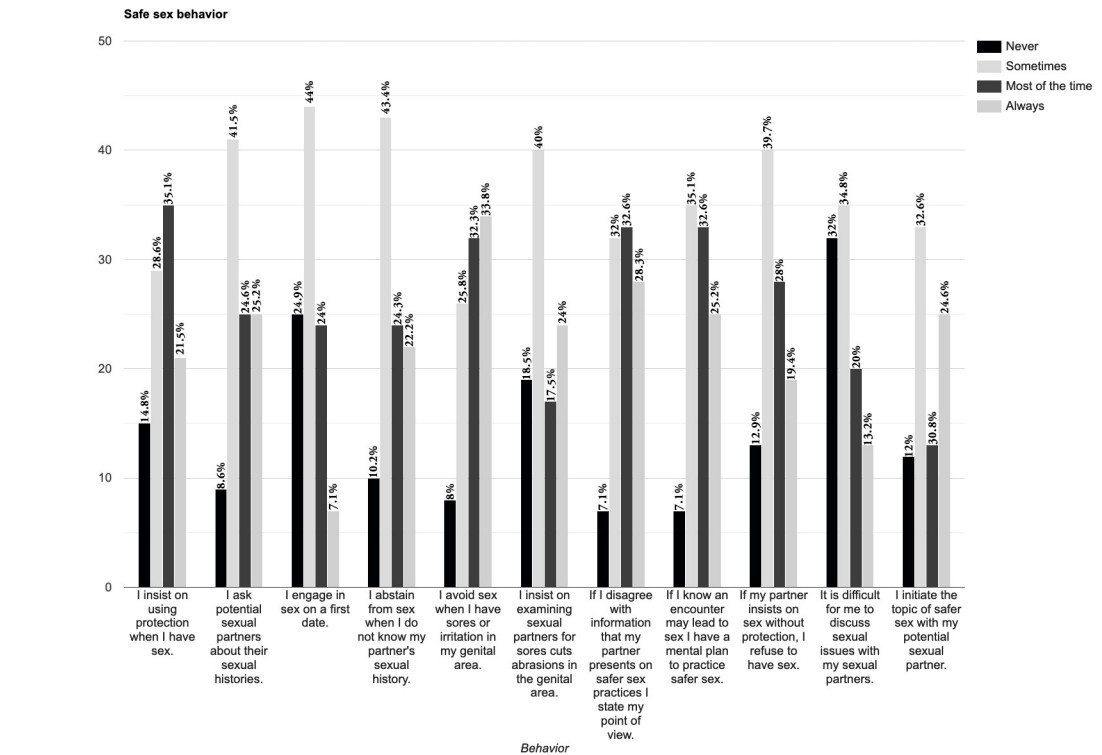

**Fig 11. Safe sex behavior questionnaire (abbreviated) (Dorio et al., 1993).**

## Risk behavior

The mean score on the Safe Sex Behavior Questionnaire was 28.3 (SD, 6.25). See Fig 11 for details. Over half (52.6%) say that they never or sometimes refuse to have sex if their partner insists on not using protection.

There was a significant statistical difference between risk behavior and religion for the group very frequent attendance and those who attend religious services least frequently compared to those (27.2 + 2.002; $p < 0.05$; CI 95% 0.29–3.71), an indication that those frequently attending services on average engage in more risky behavior and have difficulties discussing and negotiating safe sex with partners.

Overall, 28.7% (n = 94) of the participants reported using drugs or alcohol the last time they had sex. Only 25.3% (n = 83) used a condom or dental dam the last time they had sex.

## Substance use

Cigarette smoking and (binge-) drinking seemed to be common. The mean score was 12.73 (SD, 4.85). Compared to lesbian respondents, bisexual participants had statistically significantly higher scores (12.5 + 1.9; $p < 0.05$, CI 95% 0.3–3.5) and polyamorous participants had statically significantly higher scores (12.3 + 2.4; $p < 0.025$, CI 95% 0.3–4.5), indicating more use in these two groups. None of the trans men indicated taking medication without a prescription. Fig 12 gives an overview of the individual responses.

## Discussion

Our study is the first to investigate sexual health knowledge and behavior among Kenyan queer womxn and trans men. The data provide a rich and complex understanding of both

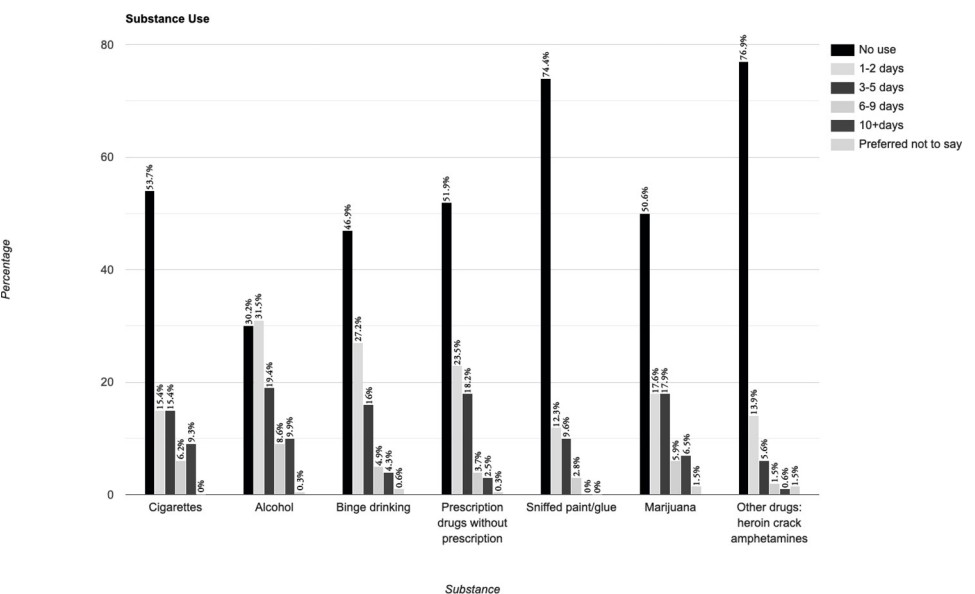

**Fig 12. Substance use over the past 30 days.**

knowledge and practices and substance use that impact this. Findings show that sexual risk behavior and substance use are prominent among Kenyan queer womxn and trans men and that there are concerns around risk-taking and preventative behavior.

## Sexual health, STDs, and behavior

The majority of survey participants (69.3%) had never used contraception. For those who had, the most used methods were male condoms (66.3%) and emergency contraception (62%, n = 62). In comparison, over half (57.4%) of Kenyan women between the ages of 15 and 49 reported no use of contraception. Injectables were the most common form of contraception (18.7% of those using contraception), followed by implants (7.1%). Male condoms only make up 3.1% of contraception methods used [39]. The measured indicators for sexual health-seeking behavior, such as PAP smears (25.7%) and HIV testing (76.1%), were comparable to Kenyan women in the general population (21.1% and 83%, respectively) [39]. At the same time, our findings suggest that, compared to the general population, queer womxn and trans men are at higher risk for STIs: One in three queer womxn and trans men in our study had been treated for an STI in the previous year. According to 2014 data, 2% of Kenyan women in the general population (assumed to be heterosexual and cisgender) reported having had an STI [39]. This is an indicator of the need for more sexual health information to prevent risky behavior and could result from the lack of commodities. It shows that queer womxn and trans men with symptoms do seek care, which is positive.

The high rate of abortions reported by queer womxn and trans men is a concern: In our sample, one in ten participants indicated having had an abortion. This corresponds to a recent study that found that 13% of Kenyan queer womxn had procured an abortion [10]. This is twice the number of abortions Kenyan women in the general population (assumed to be cisgender and heterosexual) had, according to a 2002 study [40]. Our findings point to various explanations for this: queer womxn and trans men have sex with men for a variety of reasons but may have insufficient knowledge to make informed choices around contraception. They

also experience high levels of sexual violence [41], leaving them at risk of unintended pregnancies and, subsequently, seeking illegal abortion services. Charlton et al. [42] found that, compared to heterosexual women, bisexual and queer women were as or more likely to have had a (teen) pregnancy and an abortion, while lesbian women were as likely to have had a pregnancy. The authors suggest that this is not due to risky behavior but to discrepancies in knowledge about and access to contraception. They state that this data should inform sexual health education, a critical tool for primary prevention. The legislation around abortion in Kenya is difficult to interpret, with the new Constitution of 2010 permitting abortion when the life and health of the pregnant person are in danger; however, the Penal Code has not been updated to reflect this, and unlawful abortions continue to be considered felonies, punishable with 7 to 14 years in prison. Considering the shortage of providers who perform safe and legal abortions [43], it is likely that many of the abortions that our respondents reported were procured in unsafe settings, which leaves womxn at risk for negative health outcomes.

One in four participants rated their mental health as 'fair' or 'poor,' which could impact their sexual behavior and risk taking. Several studies have associated poor mental health outcomes with sexual risk-taking, especially in gay and bisexual men. In a US study, anxiety and low self-esteem were directly related to MSM being more likely to engage in unprotected anal sex [44]. In MSM in Tanzania, research found high depression rates associated with sexual risk behavior and higher HIV incidence [45]. US trans students were more likely to be suicidal and engage in risky behavior than cis-gender students [46]. Little research is available on mental health and its effect on sexual risk-taking in queer womxn. This link between mental health status and taking sexual risks could be a concern for queer womxn and trans men and should be acknowledged when considering interventions to reduce their risk.

## Substance use

Almost 1 in 4 respondents said that they were taking hard drugs such as heroin once a month or more. Data on the substance use of the general population of Kenyan women were not available, making comparison impossible, and most existing research on substance use and risk behavior was conducted with MSM. One study found that lesbian women had twice the odds of substance use than their heterosexual peers, whereas STI-positivity was not significantly different between groups. Though this study did not report if there were significant differences between risk behavior, STI prevalence, and substance use, the authors suggest that knowing someone's sexual identity was important for understanding sexual health risk and its association with substance use [47].

Smoking of tobacco and marijuana products was a coping mechanism for depression, anxiety, and stress by Kenyan sexual minority women in a 2013 study; there was little information on hard drug and alcohol use [48]. Another study found that 47% of Kenyan lesbian respondents smoked, compared to 1% in the general female Kenyan population, and that smoking seemed negatively associated with incidences of depression [49]. In Kenyan MSM, depressive symptoms, alcohol, and substance were common. Using drugs such as methamphetamines as avoidant coping strategies has only been researched in MSM [50].

Considering that Kenyan queer womxn and trans men have to cope with many stressors in their daily lives, it seems plausible that substance use is a coping mechanism that could lead to risky sexual behavior and possibly transactional sex.

## Rape

While rape and sexual violence were not explicitly enquired about, they emerged as an area of concern. One participant indicated having been three years old at first-time sex; 14.2% were

younger than 12. This is statutory rape. The Children Act [51] considers anybody under the age of 18 a child. As per the Sexual Offences Act [52], *a person who intentionally commits rape or an indecent act (. . .) with a child (. . .) is liable upon conviction to imprisonment for a term which shall not be less than 10 years.'* Defilement (involving penetration) of a child 11 or younger leads to imprisonment for life, no less than 20 years if defiling a child between 12 and 16, and no less than 15 years for between 17 and 18 [51].

Identifying perpetrators or further consequences of rape was beyond the scope of our study. It is unclear who the perpetrators of early childhood sexual violence are or how such experiences will affect queer womxn and trans men's later sexual health, wellbeing, or relationships. While sexual violence towards young girls is unlikely to be related to sexual orientation or gender identity, research from the US suggests that lesbian and bisexual women experience more frequent, severe, and persistent levels of abuse and sexual violence during childhood compared to cisgender girls, which has been hypothesized to lead to health disparities [53]. A US study found that abuse in childhood directly predicted alcohol abuse and psychological distress in lesbian women [54]. Recent data from Kenya underscores that lesbian, bisexual and queer women are at high risk of sexual and other forms of violence [41]. Trauma-related to sexual violence, whether in childhood or adulthood, might have a detrimental effect on mental health and could be an additional stressor for Kenyan queer womxn and trans men. While there was no significant correlation between age at first time sex and substance use in this sample, more research into sexual violence, substance use, and mental health might be beneficial to understand how those factors affect the health and wellbeing of Kenyan queer womxn and trans men and to inform violence support and interventions.

## SRHR knowledge: Internet and schools are an opportunity

Sexual health is complex, with many internal and external factors affecting knowledge, attitudes, and behaviors. From the data, it emerged that there are gaps in the provision of information on SRHR issues for Kenyan queer womxn and trans men. Having intercourse with partners of various genders, sexual risk behavior, violence, and transactional sex add to the complexity of the knowledge needs queer womxn and trans men have to make healthy choices. This affects the need for more facetted education and a deeper understanding of queer womxn and trans men's sexuality that seems to be rarely provided.

Online information can bridge some of these gaps; a study from the US suggests that the lack of information on sexual health and sexuality leads young sexual minority women to find information using online resources [55]. Peer education may address knowledge gaps: receiving information from relatable people seems effective, especially for people in rural areas, who may also not have access to the internet. This is reflected in this study; the internet was named the first or second most important source of information by 43.7% of the participants. This knowledge could be important for organizations providing information for queer womxn and trans men and other sexual and gender minorities. It should be noted that information should be context-relevant and show an understanding of the lived realities of the people targeted.

For over 30% of the respondents, school was either the first or second most important source of SRHR information. This provides another opportunity, especially considering that the Kenyan government has committed to introducing CSE in schools. Currently, there are gaps: according to the Guttmacher Institute [29], only 2% of students indicate having received CSE; teachers claim 75% of schools provide it. Successfully implementing CSE does require teacher training and development of material lesson planning, suggested a comparison of European countries which had implemented CSE to those that have not [56]. In Kenya, this means that a nationwide commitment to CSE-roll-out would need to include culturally

appropriate education and sensitization for teachers and other stakeholders such as parents to ensure their alignment and improve their commitment to providing learners with CSE.

A social and human rights approach should be aimed for [57] in CSE provision. It should be inclusive and readily available, be non-judgmental and come from a place of freedom. Pleasure should be a part of it. Especially trans people are rarely included in conversations of pleasure as they tend to be reduced to their genitalia and transition conversations. Trans people have a particularly hard time finding information, but that information matters to them in the first place.

## Limitations

The main limitation of this study is the sample size of 335 participants, and the methodology of sampling, recruiting and collecting data online introduced bias. This bias needs to be considered when interpreting the results. As very little is known about the target population's demographic profile in Kenya, it was not possible to determine whether the participants were a true representation of the population. The present sample represents young and highly educated queer womxn and trans men, and these findings are hence unlikely to be representative. However, the intention was to surface key issues in a hard-to-reach, under-researched population. Future research should ensure that a broader sample is included to accurately assess the needs of all Kenyan queer womxn and trans men.

Secondly, topics around sexual health are complex–shaped by many factors beyond the scope of this research. Further research will need to address various topics around knowledge, attitudes, and behavior in more depth than was possible for this study. For example, the questioning around transactional sex did not explore how people engaged with it, whether they provided or received it. Disaggregating this would help address any concerns and risks around transactional sex. Additionally, the authors are aware that social desirability in sexual health research can affect data collection outcomes.

Due to the small sample sizes of groups such as GNC (n = 29) and trans men (n = 27) participants, more in-depth statistical analyses such as multivariate regression models could not be conducted as the power would have been very low.

The involvement of community members positively contributed to the data collection strategy; however, queer womxn and trans men not associated with the networks in which the messages about the survey were spread were unlikely to hear about the research and were hence unable to contribute. It is possible that these people have a different experience around sexual health and service use.

Even though measures were put in place, it should be noted that multiple participations could not be fully eliminated with a self-administered online survey tool.

Finally, constructs around culture, policy, and religion are varied, multi-layered, and deeply rooted, and more detailed studies and instruments will be needed to get a comprehensive, even more in-depth understanding of how they influence sexual health behavior.

## Conclusion

Our findings indicate that queer womxn and trans men in Kenya are at risk of negative sexual health outcomes due to a lack of appropriate information, risk behavior, and substance use. Kenya's Penal Code still criminalizing consensual same-sex activities may play a significant role in perpetuating the barriers that prohibit queer womxn and trans men from making healthier choices; the same is true for other countries with legislation that restricts consensual same-sex activities. Developing more tailored, effective programming and policies require local, national, and global stakeholders to acknowledge and actively engage with the inclusion of queer womxn and trans men's sexual health needs and risk behaviors within strategic planning and delivery to improve

health outcomes and potentially overall burden of disease. The structural and social contexts in which queer womxn and trans men live can endanger their health and wellbeing and decrease their access to information and services to improve sexual health. This places queer womxn and trans men in double jeopardy. Within restrictive legal contexts, these risks might be amplified through social exclusion. Research, programming, and information addressing sexual health need to include queer womxn and trans men explicitly and acknowledge the factors and barriers that shape their sexual health and risks to address gaps and areas of concern.

### Research

Moving forward, queer womxn and trans men should explicitly be included in research around the needs of sexual and gender minorities in order to be able to adequately address their needs from a public health and human rights perspective. More generally, data on SOGIE should be collected in research such as Demographic and Health Surveys so that SOGIE-specific health disparities and needs can be identified.

### Programming and information provision

The knowledge generated from research for and with queer womxn and trans men should be used to inform choices and planning in programming, service delivery, and information provision to ensure the inclusion of this population and meet their unique needs within the greater public health system.

## Acknowledgments

The authors would like to thank the research participants and all other collaborators for their important contribution to the study. I (SH) would also like to thank my supervisors from the University of Cape Town.

## Author Contributions

**Conceptualization:** Stephanie Haase, Alex Müller, Virginia Zweigenthal.

**Data curation:** Stephanie Haase.

**Formal analysis:** Stephanie Haase.

**Methodology:** Stephanie Haase.

**Software:** Stephanie Haase.

**Supervision:** Alex Müller, Virginia Zweigenthal.

**Visualization:** Stephanie Haase.

**Writing – original draft:** Stephanie Haase.

**Writing – review & editing:** Alex Müller, Virginia Zweigenthal.

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
