## [Decision Letter · Decision Letter 0]

26 Jan 2022

PONE-D-21-30256Sexual health behavior, health status, and knowledge among queer womxn and trans men in Kenya: an online cross-sectional studyPLOS ONE

Thank you for submitting your manuscript to PLOS ONE. After careful consideration, we feel that it has merit but does not fully meet PLOS ONE’s publication criteria as it currently stands. Therefore, we invite you to submit a revised version of the manuscript that addresses the points raised during the review process.

We look forward to receiving your revised manuscript.

Kind regards,

Luigi Lavorgna

Academic Editor

PLOS ONE

Journal Requirements:

Unfunded study

NO authors have competing interests

Reviewers' comments:

Reviewer's Responses to Questions

**Comments to the Author**

1. Is the manuscript technically sound, and do the data support the conclusions?

Reviewer #1: Yes

2. Has the statistical analysis been performed appropriately and rigorously? 

Reviewer #1: Yes

3. Have the authors made all data underlying the findings in their manuscript fully available?

Reviewer #1: Yes

4. Is the manuscript presented in an intelligible fashion and written in standard English?

Reviewer #1: Yes

5. Review Comments to the Author

Reviewer #1: Haase and colleagues reported on sexual health behavior, health status, and knowledge among queer womxn and trans men in Kenya (n=335), through an online survey shared by whatsapp by influencers of the community. The manuscript is overall clear and well written. Methods are sufficiently sound. The topic is interesting, especially considering the inclusion of an understudied population in Africa and the fact that Kenya's Penal Code still criminalizes consensual same-sex activities. I only have some minor comments to the authors.

In the introduction, when referring to the healthcare impact of queer womxn and trans men, authors might want to refer to queer womxn and trans men, showing that sexual orientation actually affects healthcare resource utilization and health status also in countries not criminalizing consensual same-sex activities. As such, this study is also relevant in terms of cultural change.

Table 3. Could you please add percent of participants in the new grouping column?

By any chance, have authors investigated race/ethnicity of participants?

“The majority, 69.3% (n=232), had never used any form of contraception”. Do you have any corresponding figure from the general population in Kenya?

I really liked the use of community influencers for sharing the survey in a country that criminalizes consensual same-sex activities. While I agree influencers play a very important part in communities for sharing reliable information (e.g., 10.1016/j.msard.2018.07.046), I also believe this should be discussed among limitations with the inclusion of a subset of the queer womxn and trans men community (e.g., in-closeted people do not belonging to the community were not reached).

6. PLOS authors have the option to publish the peer review history of their article (what does this mean?). If published, this will include your full peer review and any attached files.

Reviewer #1: No

---

## [Author Response · Author response to Decision Letter 0]

6 Apr 2022

Dear Reviewer, 

Thank you very much for your kind words and productive feedback. 

Please find below our responses. 

1. In the introduction, when referring to the healthcare impact of queer womxn and trans men, authors might want to refer to queer womxn and trans men, showing that sexual orientation actually affects healthcare resource utilization and health status also in countries not criminalizing consensual same-sex activities. As such, this study is also relevant in terms of cultural change.

Thank you very much for this very relevant point. This was added on page 5, line 94 onward. 

2. Table 3. Could you please add percent of participants in the new grouping column?

These have now been added, thank you. 

3. By any chance, have authors investigated race/ethnicity of participants?

Unfortunately, we did not. 

4. “The majority, 69.3% (n=232), had never used any form of contraception”. Do you have any corresponding figure from the general population in Kenya?

Indeed, there are data! Thank you for suggesting adding this, which was done on page 18, line 356-362. 

5. I really liked the use of community influencers for sharing the survey in a country that criminalizes consensual same-sex activities. While I agree influencers play a very important part in communities for sharing reliable information (e.g., 10.1016/j.msard.2018.07.046), I also believe this should be discussed among limitations with the inclusion of a subset of the queer womxn and trans men community (e.g., in-closeted people do not belonging to the community were not reached).

Thank you, we agree with this assessment and have added this to the limitations, page 24, line 514 t0 518. 

Once again, we appreciate the time you took to review our manuscript and the kindness of your feedback.

---

## [Editor Report · Decision Letter 1]

27 Apr 2022

Sexual health behavior, health status, and knowledge among queer womxn and trans men in Kenya: an online cross-sectional study

PONE-D-21-30256R1

We’re pleased to inform you that your manuscript has been judged scientifically suitable for publication and will be formally accepted for publication once it meets all outstanding technical requirements.

Kind regards,

Luigi Lavorgna

Academic Editor

PLOS ONE

Additional Editor Comments (optional):

add the reference indicated by the reviewer
---

## [Editor Report · Acceptance letter]

26 May 2022

PONE-D-21-30256R1 

Sexual health behavior, health status, and knowledge among queer womxn and trans men in Kenya: an online cross-sectional study 

Dear Dr. Haase:

I'm pleased to inform you that your manuscript has been deemed suitable for publication in PLOS ONE. Congratulations! Your manuscript is now with our production department. 

Kind regards, 

on behalf of

Dr. Luigi Lavorgna 

Academic Editor

PLOS ONE